# Basal melt of the southern Filchner Ice Shelf, Antarctica

Ole Zeising[1,2], Daniel Steinhage[1], Keith W. Nicholls[3], Hugh F. J. Corr[3], Craig L. Stewart[4,5], and Angelika Humbert[1,2]

[1]Alfred-Wegener-Institut Helmholtz-Zentrum für Polar- und Meeresforschung, Bremerhaven, Germany
[2]Department of Geosciences, University of Bremen, Bremen, Germany
[3]British Antarctic Survey, Natural Environment Research Council, Cambridge, UK
[4]Scott Polar Research Institute, University of Cambridge, Cambridge, UK
[5]now at: National Institute of Water and Atmospheric Research, Wellington, New Zealand

**Correspondence:** Ole Zeising (ole.zeising@awi.de)

**Abstract.** Basal melt of ice shelves is a key factor governing discharge of ice from the Antarctic Ice Sheet as a result of its effects on buttressing. Here, we use radio echo sounding to determine the spatial variability of the basal melt rate of the southern Filchner Ice Shelf, Antarctica along the inflow of Support Force Glacier. We find moderate melt rates with a maximum of $1.13\,\mathrm{m\,a^{-1}}$ about 50 km downstream of the grounding line. The variability of the melt rates over distances of a few kilometres is low (all but one $< 0.15\,\mathrm{m\,a^{-1}}$ at $< 2\,\mathrm{km}$ distance), indicating that measurements on coarse observational grids are able to yield a representative melt rate distribution. A comparison with remote sensing based melt rates revealed that, for the study area, large differences were due to inaccuracies in the estimation of vertical strain rates from remote sensing velocity fields. These inaccuracies can be overcome by using modern velocity fields.

## 1 Introduction

Filchner Ice Shelf (FIS), a West Antarctic ice shelf draining major East Antarctic ice streams (Bailey, Slessor, Recovery and Support Force glaciers) is thought to be vulnerable to a change in its basal mass balance within this century (Hellmer et al., 2012) as a result of the possible penetration of relatively warm, off-shelf waters into the ocean cavity beneath the ice shelf. Subsequent thinning of the ice shelf would reduce its buttressing to inland glaciers, allowing them to speed up and thin, and their grounding lines to retreat landward. If the stress perturbation is sufficiently large then a positive ice-loss feedback may occur as the ice sheet's grounding line retreats across the deepening beds of the tributary ice streams (Schoof, 2012). The current discharge of ice across the grounding line at FIS is $106.3 \pm 5.7\,\mathrm{Gt\,a^{-1}}$ (Rignot et al., 2019), which is about $9.6\,\%$ of the discharge from East Antarctica, underlining the importance of understanding the current state of the ice shelf for assessing future change in basal melt. In addition, precise melt rates serve as validation for models projecting the future contribution of these ice streams to sea level change.

Basal melt rates can be derived from remote sensing data by solving the ice thickness evolution equation (Rignot et al., 2013; Moholdt et al., 2015; Berger et al., 2017; Adusumilli et al., 2020). Although the Lagrangian approach adopted in recent years (Moholdt et al., 2015) has led to improvements, major uncertainties from various factors remain. Hence, in situ observations of basal melt rates are required for assessing the reliability of remote sensing approaches. This is even more urgent, as remote

sensing-derived basal melt rates are used to construct parametrisations that diagnose basal melt rates from modelled sub-ice shelf ocean conditions. These models are used to project the contribution of Antarctica to sea level change. Significant errors in observed distributions of basal melt rate therefore have a profound effect on the outcome of projections of future sea level rise, such as ISMIP6 (Seroussi et al., 2020), as a result of their effect on the calibration of basal melt rate parametrisations (Jourdain et al., 2020).

In recent years, the use of the phase-sensitive radio echo sounder (pRES) opened new possibilities for the precise determination of basal melt rates. Nicholls et al. (2015) and Stewart et al. (2019) presented basal melt rates from near Ross Island, Ross Ice Shelf, Antarctica, which were derived from 10-days of autonomous pRES (ApRES) measurements, and measurements from 78 stations, time-averaged between 2013 and 2014. Stewart et al. (2019) observed strong seasonal melt rate variability, with values up to $53\,\mathrm{m\,a^{-1}}$ within a five day period in January 2013 and an exponentially reducing mean annual basal melt rate with increasing distance from the calving front, with values up to $7.7\,\mathrm{m\,a^{-1}}$. Vaňková et al. (2020) presented a tidal melt and vertical strain analysis from 17 ApRES records across Filchner-Ronne Ice Shelf. They found the tidal vertical strain to be depth dependent only near the grounding line, with significant tidal melt measurable at some locations. The derived melt rates were used by Bull et al. (2021) to evaluate an ocean model. Marsh et al. (2016) investigated basal melt rates at 25 points at a melt channel near the grounding line of Ross Ice Shelf. They found basal melt rates decreasing from $22\,\mathrm{m\,a^{-1}}$ at the upstream end of the channel to $2.5\,\mathrm{m\,a^{-1}}$ 40 km downstream. A strong seasonal variability in melt rate was recorded by Washam et al. (2019) on Petermann Gletscher, Greenland, using an ApRES recording on the flank of a basal melt channel. In Summer 2016, they found extreme melt rates equivalent to $80\,\mathrm{m\,a^{-1}}$ but most of the year the mean basal melt rate ranged from 0 to $10\,\mathrm{m\,a^{-1}}$.

Our survey is focused on the accessible southern part of FIS, which might be more susceptible to the potential inflow of warm waters (Hellmer et al., 2012). Recent observations from hot-water drilled boreholes through Filchner-Ronne Ice Shelf have revealed an interannual change in circulation mode starting in 2017, highlighting the variability in conditions within the sub-ice shelf cavity (Hattermann et al., 2021).

Here, we aim at understanding the magnitude and variation of basal melt over an area extending from the grounding line of Support Force Glacier, as far downstream as was feasible. In austral summer 2015/16, under the framework of the Filchner Ice Shelf Project (FISP), pRES measurements were carried out at a total of 94 locations, and then repeated a year later. The stations were distributed along the central flow line of Support Force Glacier's extension on to FIS and along four cross-sections, providing along-flow and across-flow melt rate distributions (Fig. 1). A further transect crossed the entire FIS south of Berkner Island. As far as safety allowed, we extended the profiles along the eastern margin towards the inland ice, to capture an area where gradients in the bathymetry were expected, steering the flow of water masses. With this observational design we intended to measure the large scale distribution of melt rates, but in addition we included more closely spaced stations to detect variations on short spatial scales. In the following, we first introduce the methodology and the data basis. We then present and discuss the derived basal melt rates and compare them with remote sensing data.

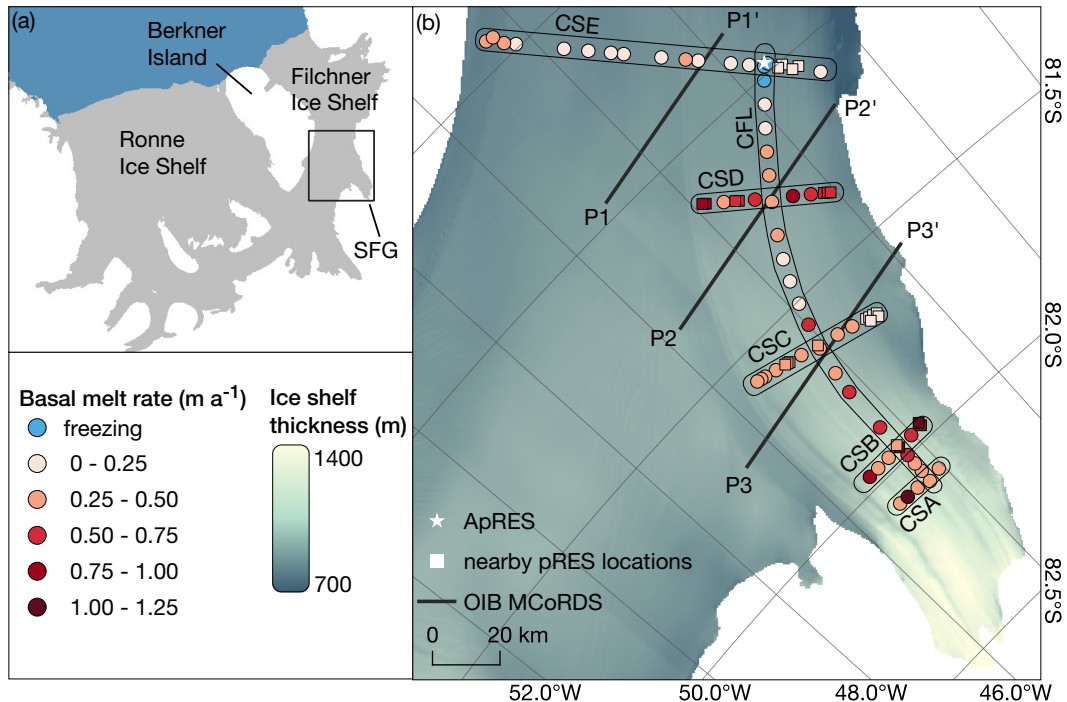

**Figure 1.** (a) Map of the Ronne and Filchner ice shelves (BedMachine Antarctica (Morlighem, 2020; Morlighem et al., 2020)) with the marked study area (black box) near the Support Force Glacier (SFG). (b) Study area with derived basal melt rates (dots), grouped depending on their location on the Central Flow line (CFL) and five Cross-Sections CSA, CSB, CSC, CSD and CSE. Nearby pRES measurements with distances < 2 km are shown by squares and the location of an ApRES station (FSE1 in Vaňková et al., 2020) is shown by a star. Multi-channel Coherent Radar Depth Sounder (MCoRDS) profiles (P1–P3) flown as part of NASA's Operation IceBridge (OIB) campaign in 2016 (Paden et al., 2014, updated 2019) are shown by black lines (Echograms are included in the Appendix, Fig. A1). Background colour shows the ice-shelf thickness from BedMachine Antarctica (Morlighem, 2020; Morlighem et al., 2020).

## 2  Materials and methods

Our estimation of basal melt rates is based on measurements using a pRES that is described in detail in Brennan et al. (2014) and Nicholls et al. (2015). The pRES transmits a frequency modulated sweep (chirp) from 200 to 400 MHz over a period of one second via two skeleton slot antennas, separated by roughly 9 m. The exact locations were marked with bamboos for precise
relocation a year later. After internal processing, only the difference in frequency between the transmitted and received signals, called the deramped frequency, is saved. Details of the internal processing are given by Brennan et al. (2014). By repeating the measurements after a time period, we are able to track changes in depth of internal reflectors within the ice, and of the basal echo, to a precision of millimetres. This allows the study of firn densification, vertical strain due to ice flow, and the (Lagrangian) change in ice-shelf thickness. Being a Lagrangian measurement, no steady state assumption is required, and the
basal melt rate can be separated from the overall change in ice thickness.

Our 94 measurement stations are grouped depending on their locations on the Central Flow line (CFL) and five Cross-Sections (CS) A – E (for location, see Fig. 1). The time period between repeated measurements varied from 323 (18 Jan 2016 – 06 Dec 2016) to 356 days (31 Dec 2015 – 21 Dec 2016). To improve the signal-to-noise ratio, we recorded 100 chirps at each site. Correlations were calculated between each chirp and the 99 remaining chirps, and those chirps with a low average correlation were discarded. Those remaining chirps were averaged and then Fourier transformed to yield a complex (amplitude and phase) profile as a function of two-way-travel time. To convert the profile into a function of range we calculated the velocity profile of the electromagnetic wave for each location by estimating the density-depth profile based on Herron and Langway (1980) with accumulation and mean annual temperature from RACMO 2.3/ANT (van Wessem et al., 2014). Nevertheless, the uncertainty of the propagation velocity is about $1\%$ (Fujita et al., 2000).

Using a procedure similar to that described by Corr et al. (2002) and Jenkins et al. (2006), we aligned the two radar profiles using a $6\,\mathrm{m}$ window below the firn-ice boundary by cross-correlating the amplitude profiles. This provided a datum within the ice column, removing the effects of instrument temperature change, firn densification and snow accumulation and ablation.

The thickness change ($\mathrm{D}H_i/\mathrm{D}t$) in the solid-ice below the aligned reflector is caused only by the dynamic ice thickness change due to vertical strain ($H_i\dot{\varepsilon}_{zz}$) and by the basal melt rate $a_b$:

$$\frac{\mathrm{D}H_i}{\mathrm{D}t} = H_i\dot{\varepsilon}_{zz} - a_b, \tag{1}$$

with $H_i$ the solid-ice thickness below the aligned reflector and $\dot{\varepsilon}_{zz}$ the vertical strain rate. In order to determine the vertical strain, the displacement between visits was calculated with a cross-correlation of the amplitude and phase information for each layer deeper than the aligned reflector. Under the plane-strain assumption the vertical strain is constant with depth; a least-squares method was used to calculate a linear fit of the shift of those layers that exhibited a high correlation value. The gradient of the linear fit is the vertical strain. The change in ice thickness below the aligned reflector is derived from the shift of the basal reflector, which was calculated in the same way as the shift of the internal layers. The largest error in the calculation comes from the alignment of the data because it is based only on the amplitude correlation. The uncertainty in the calculation of the phase shift is closely related to the signal-to-noise ratio of the reflectors. An additional uncertainty arises from the assumption of a linear displacement-depth relation, although this is generally thought reliable for plug flow (Greve and Blatter, 2009). These uncertainties add up to $0.03\,\mathrm{m}$.

## 3 Results and discussion

### 3.1 Large scale spatial variability

Seventy-nine of the 94 measurements were suitable for retrieval of basal melt rates. The main reasons for excluding the other 15 stations are (1) low correlation values in the depth of the firn-ice transition, which made it impossible to align the measurements, (2) changes in the shape of the basal reflector that prevented the reflections from being unequivocally matched, (3) too few high correlation values for a linear fit to be used to calculate the vertical strain rate. An explanation for the low correlation values could be errors in operating the pRES, such as inaccuracies in the alignment of the antennas or incorrectly seated cables. In

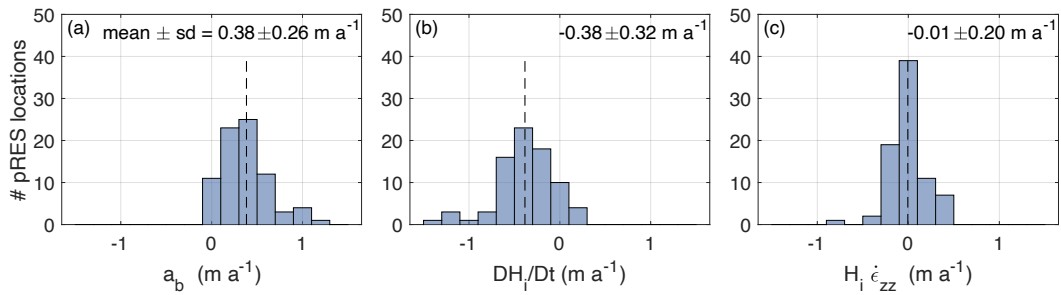

**Figure 2.** Distribution of pRES-derived results of (a) the basal melt rate $a_b$, (b) the change in solid-ice thickness $DH_i/Dt$ and (c) the dynamic change in solid-ice thickness $H_i\dot{\varepsilon}_{zz}$. The numbers in the upper right corner state the mean value and the standard deviation (sd).

addition, changes in settings such as the attenuation affected the signal-to-noise ratio, thereby reducing the number of high correlation values.

For the remaining stations, we found a mean basal melt rate of $0.38 \pm 0.26 \, \mathrm{m\,a^{-1}}$ (mean + standard deviation; Fig. 2a) with a maximum of $1.13 \pm 0.03 \, \mathrm{m\,a^{-1}}$ at a location about $50 \, \mathrm{km}$ downstream of the grounding line and freezing in the northernmost part of the central flow line (CFL). Of similar size but with different sign, the mean value of $DH_i/Dt$ is $-0.38 \pm 0.32 \, \mathrm{m\,a^{-1}}$ (Fig. 2b), representing a thinning of the solid ice, whereas the mean value of $H_i\dot{\varepsilon}_{zz}$ ($-0.01 \pm 0.20 \, \mathrm{m\,a^{-1}}$) is close to zero (Fig. 2c).

This study focuses on the spatial variability of the melt rates, rather than the overall, annual average values, since we did not measure the interannual variability. However, the different sites were occupied for different periods and thus, seasonality in basal melt would affect spatial variability if it exists. The time periods are ranging from 365 minus 42 days to 365 minus 9 days with increasing data acquisition interval southwards. Seasonality may affect the derived annual melt rate differently at the different sites. However, in the northern part of our study area an autonomous pRES (ApRES) station recorded for more

than a year, including the period of the single-repeated pRES measurements (Fig. 1b, Vaňková et al., 2020). For the time period between 18 Jan 2016 and 06 Dec 2016 (same period as the pRES measurements with the shortest time interval), we derived a melt rate of $0.02 \pm 0.03 \, \mathrm{m\,a^{-1}}$. In the period of 365 days (18 Jan 2016 – 17 Jan 2017), a slightly lower melt rate of $0.01 \pm 0.03 \, \mathrm{m\,a^{-1}}$ was derived. This indicates that no enhanced melting occurred at the location of the ApRES in summer 2016/2017. However, we cannot assess if melt rates are enhanced at other locations.

We present the distribution of $a_b$ in Fig. 1b, as well as an along-flow profile (CFL) and five cross-sections (CSA-CSE) in Fig. 3. Seventy percent of the estimated basal melt rates range between $0$ and $0.50 \, \mathrm{m\,a^{-1}}$. Higher melt rates were found for nine stations within $100 \, \mathrm{km}$ of the grounding line at the CFL, CSA and CSB. All three stations with $a_b > 1 \, \mathrm{m\,a^{-1}}$ are located in this part of the study area. The variation of $a_b$ along ice flow is weak and shows no clear trend of increasing melt towards the grounding line, despite the increasing ice draft (Fig. 3a). In the direction across ice flow (Fig. 3b-f) the largest variations

in $a_b$ appear in the two southernmost cross-sections (CSA, CSB). Apart from the southern part, higher basal melt rates of up to $0.82 \pm 0.03 \, \mathrm{m\,a^{-1}}$ occur only at CSD. The northernmost cross-section, ranging from Berkner Island towards the inland ice

(CSE), has a generally low $a_b$. Three stations, all at the northernmost part of CFL, indicate freezing.

A key assumption made during pRES processing is that the phase-shift on reflection at the ice-ocean interface remains constant. Although this is valid for a fresh ice/seawater interface, it is a poor assumption for the interface between fresh ice and possibly slushy marine ice, itself underlain by seawater. This means that if either of a pair of measurements is made during a period of freeze-on it is not possible to distinguish the change in the phase of the basal reflection that results from a change in its range, from the phase shift that results from the change in the nature of the basal interface. Thus, we can not determine the amount of accretion at the three sites.

In an ice shelf cavity where the water speeds are relatively high as a result either of tides, as in this case (Mueller et al., 2018), or as a result of strong buoyancy-driven flows, as in the case of a steeply-inclined ice base over relatively warm water (Lazeroms et al., 2019), basal melt rates are mainly controlled by three factors: the basal drag coefficient, the thermal driving, and the water speed in the boundary layer. The thermal driving is the difference in the temperature of the water near the ice base and the freezing point of that water at the pressure of the ice base. The water speed and the basal drag generate the shear-driven turbulence that efficiently diffuses heat and salt towards the ice base (Holland and Jenkins, 1999). A fourth factor, discussed below, is the basal vertical temperature gradient in the ice.

In our study area the basal slopes are generally low (Morlighem, 2020; Morlighem et al., 2020), as is the thermal driving. We therefore expect tidal speeds to dominate buoyancy-driven flows. Ice draft plays the role of modifying the thermal driving: lower basal pressures reduce the freezing point, thereby reducing the local freezing temperature. Mueller et al. (2018) find a strong increase in tidal speeds from the grounding line to the Cross-Section CSE; this parallels a reducing basal draft, which will act to reduce the thermal driving. We expect these two tendencies to work together to modulate the large scale spatial variation in basal melt rates.

The large scale spatial variation in $a_b$ can also be influenced by changes in vertical gradients of the ice temperature. An ice shelf fed by a fast glacier typically contains a cold core as a result of ice advection, leading to larger vertical temperature gradients some distance from the grounding line. However, with melting over centuries, the ice temperature is more likely to approach a parabolic profile, with only moderate temperature gradients (Humbert, 2010).

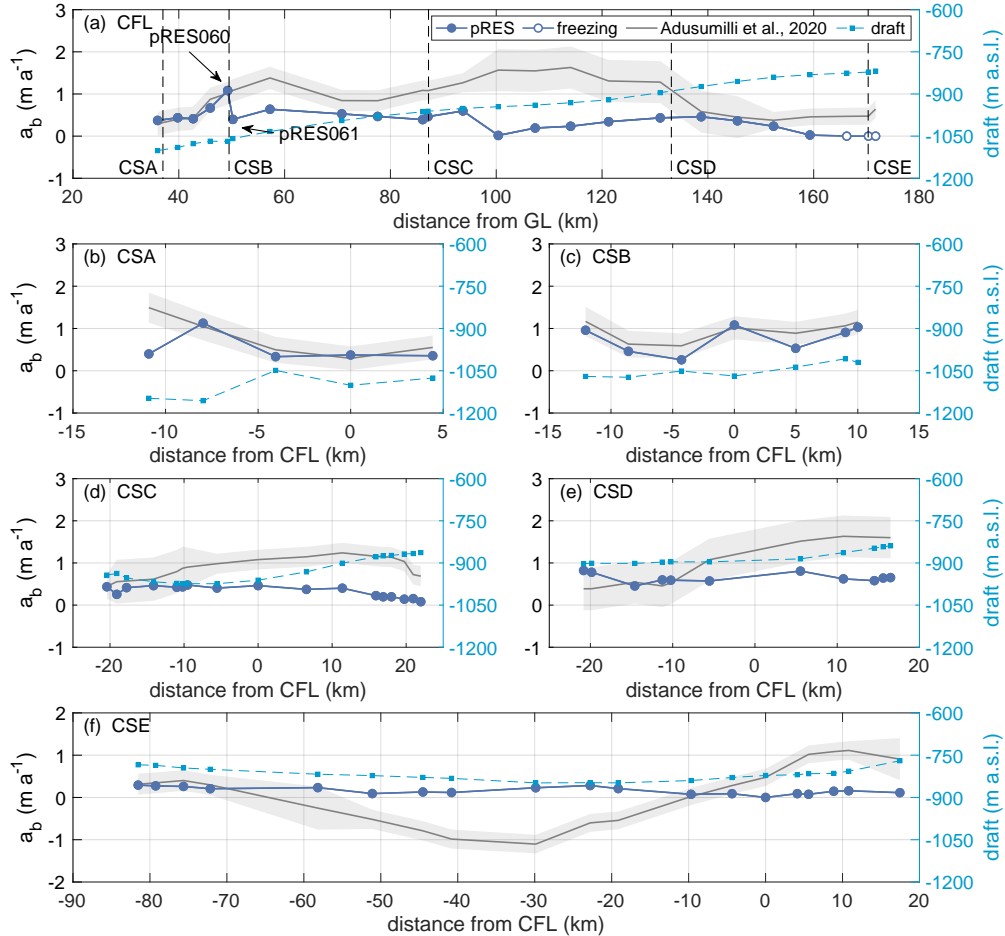

**Figure 3.** Variation of the basal melt rate (a) along the Central Flow Line (CFL) of Support Force Glacier's extension on the FIS and (b – f) the cross-sections CSA – CSE (left y-axis). Locations are shown in Fig. 1. pRES-derived values are shown in dark blue. The dark grey line represents remote sensing-derived melt rates and the light grey bounds display the uncertainty, both published by Adusumilli et al. (2020). Uncertainties of the pRES derived melt rates are $0.03\,\mathrm{cm}$ and therefore too small to visualise. The variation of the ice draft (in meter above sea level) is shown on the right y-axis. For CFL, the distance refers to the grounding line (GL) of Support Force Glacier and for all cross-sections to the CFL with positive distances on the eastern side.

## 3.2 Small scale spatial variability

In order to assess the small scale spatial variability of the basal melt rates and hence the representativeness of our point measurements over larger distances, we carried out 17 pRES measurements that were located within 2 km of another measurement (Fig. 1).

Overall, the local differences in melt rate ($\Delta a_b$) are small (Fig. 4). Except for one outlier, $\Delta a_b$ ranges from 0 to $0.13\,\mathrm{m\,a^{-1}}$ with a median of only $0.02\,\mathrm{m\,a^{-1}}$. In the case of the outlier station pair (pRES060 and pRES061), the difference was $0.68 \pm 0.06\,\mathrm{m\,a^{-1}}$ within a distance of $977\,\mathrm{m}$ (Fig. 3 and Fig. B1). Here, the higher melt rate was measured at site pRES060, which has a $10.5\,\mathrm{m}$ higher ice-shelf draft, as derived from the surface elevation given by BedMachine (Morlighem, 2020; Morlighem et al., 2020) and the pRES-derived ice thickness.

Variability at small spatial scales will not result from variations in tidal speed: in the absence of strong sea floor or ice base topography, strong horizontal gradients in tidal speed are not expected. As previously discussed, tides will dominate buoyancy-driven currents, which are therefore also unlikely to play a significant role in controlling local variations in melt rate. However, through its effect on thermal driving even a quite modest local variation in basal depth is a candidate for causing variation in basal melting. A change in ice draft of, say, $10\,\mathrm{m}$ will change the thermal driving by about $0.007\,^{\circ}\mathrm{C}$ (e.g. Holland and Jenkins, 1999). Using the algorithm proposed by Jenkins et al. (2010) for the nearby Ronne Ice Shelf, and for an appropriate mean tidal speed of $0.1\,\mathrm{m\,s^{-1}}$ (Mueller et al., 2018), a $0.007\,^{\circ}\mathrm{C}$ change in thermal driving would result in a melt rate difference of $0.17\,\mathrm{m\,a^{-1}}$. Given thermal stratification in the underlying water column, it is also possible for a locally increased draft to result in the ice base having contact with warmer waters, leading to a local increase in melt rates. Since such an increased draft of $10.5\,\mathrm{m}$ was found at pRES060 compared to pRES061, this could be a possible explanation for the large difference in melt rate of $0.68 \pm 0.06\,\mathrm{m\,a^{-1}}$. Another candidate driver of local variability in melt rate is spatial variability in basal roughness. Differences in the drag coefficient at the ice base will directly affect melt rates (e.g. Holland and Jenkins, 1999). However, neither the ice roughness nor its spatial variation is known well enough to determine its importance in driving local spatial variation in melt rates.

Overall, this analysis gives evidence that individual measurements are representative of a large area on the scale of 1–3 ice thicknesses. Only minor variation is to be expected due to the specific choice of measurement location. Airborne radar echograms (Fig. A1), recorded within NASA's Operation IceBridge (OIB) with a Center for Remote Sensing of Ice Sheets (CReSIS) Multi-channel Coherent Radar Depth Sounder (MCoRDS) in 2016 (Paden et al., 2014, updated 2019), show that our study area is mostly characterised by a smooth ice shelf base without terrace structures, supporting our interpretation that small-scale variability in basal melt rate is relatively low. One exception is a basal channel in the west with a height of approx. $50\,\mathrm{m}$ (Fig. A1e,f). At three locations around this channel, pRES measurements have been performed (western part of CSC in Fig. 3d). These show low variability, with a basal melt rate $\sim 0.2\,\mathrm{m\,a^{-1}}$ lower in the centre of the channel.

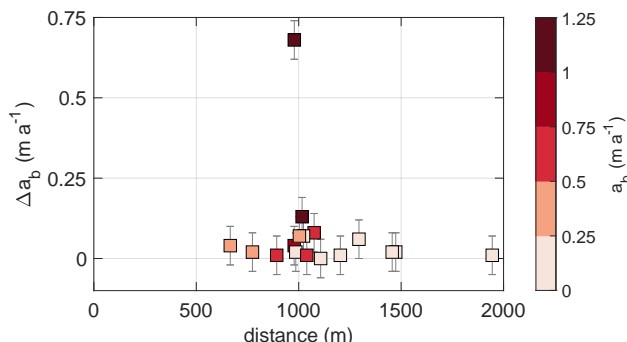

**Figure 4.** Small scale variability of basal melt rates. Difference in basal melt rate $\Delta a_b$ as function of distance between two nearby stations (average distance: $1126 \pm 296$ m). The colour of each dot represents $a_b$ of the station with the larger basal melt rate.

## 4 Comparison with remote sensing

The analysis of remote sensing-derived basal melt rates is based on precisely measured elevation changes of the ice-shelf surface and on the correction of the surface mass balance, firn densification and dynamic change in ice thickness (e.g. Moholdt et al., 2015; Adusumilli et al., 2020). The dynamic change in ice thickness and thus the vertical strain rate is often derived from the divergence of a satellite sensor-derived surface velocity field.

We used the pRES-derived vertical strain rates to assess the reliability of strain rates derived from different remote sensing velocity fields. Satellite-derived melt rates at FIS from Rignot et al. (2013), Moholdt et al. (2015) and Adusumilli et al. (2020) were all based on the strain rates derived from the same early MEaSUREs velocities (Rignot et al., 2011; Scheuchl et al., 2012). However, this velocity field contained some significant data gaps in our study area that were not present in modern velocity fields such as the Landsat Ice Speed of Antarctica (LISA) product from which vertical strain rates were derived by Alley et al. (2018) or the newest MEaSUREs data set (Mouginot et al., 2019a, b). Instead of comparing the vertical strain rate itself, we compared the dynamic ice thickness change ($H_i \dot{\varepsilon}_{zz}$) that was derived from the vertical strain rate and the solid-ice thickness. The result reveal a significant improvement over the last decade in the accuracy of the determination of vertical strain rates from remote sensing.

While the average deviation between the pRES-derived product and that from Moholdt et al. (2015) was $0.40 \pm 0.44\,\mathrm{m\,a^{-1}}$ (mean $\pm$ standard deviation; Fig. 5a,b and Fig. C1), there were much smaller deviations ($-0.01 \pm 0.35\,\mathrm{m\,a^{-1}}$) from the product of Alley et al. (2018) (Fig. 5c,d and Fig. C1). The comparison with the dynamic ice thickness that we calculated using the latest MEaSUREs data set (Mouginot et al., 2019a, b) also showed only minor deviations of $0.04 \pm 0.17\,\mathrm{m\,a^{-1}}$ (Fig. 5e,f and Fig. C1). Here, similar to Moholdt et al. (2015), we applied a Gaussian filter with a $27 \times 27\,\mathrm{km}$ window to smooth the velocity data, and calculated the divergence to obtain the vertical strain rate. The comparison highlights the recent improvement in the estimation of velocity fields for more accurate calculation of dynamic ice thickness changes, and demonstrates good agreement between remote sensing-derived strain rates and those from in situ measurements.

Remote sensing-derived melt rates published by Rignot et al. (2013), Moholdt et al. (2015) and Adusumilli et al. (2020) suggested a similar pattern of melt rates: southeast of Berkner Island, a freezing regime in the west switches to a melting regime eastwards, with melting persisting towards the south to the Support Force Glacier. However, a data gap in the velocity field meant that no melt rates could be determined by Rignot et al. (2013) for a large part of our study area.

The comparison with the results from Adusumilli et al. (2020) reveals a broader distribution of the remote sensing-derived melt rate ($-1.1 - 1.6\,\mathrm{m\,a^{-1}}$) at the pRES locations with an average deviation from the pRES-derived values of $0.35 \pm 0.57\,\mathrm{m\,a^{-1}}$ (Figs. 3 and 5g,h), which is of size similar to the deviation of the dynamic change in ice thickness. Another reason for the discrepancies can be the different measurement periods over which the basal melt rates were estimated: Adusumilli et al. (2020) shows that basal melt rates can vary at interannual timescales. In order to investigate whether different measurement periods contributed to the discrepancies between the results from the different methods, we compared the change in ice thickness $\mathrm{D}H_i/\mathrm{D}t$ (Eq. 1) after the correction for the surface mass balance and firn densification (Fig. C2). Some of the differences occur because Adusumilli et al. (2020) defines $H_i$ as the ice-shelf thickness in units of m of ice equivalent, which is slightly higher than the solid-ice thickness that we use for the pRES-based estimates. However, the comparison of $\mathrm{D}H_i/\mathrm{D}t$ shows a good agreement, with an average difference of only $0.04 \pm 0.24\,\mathrm{m\,a^{-1}}$ (Fig. 5i,j). Since variations in basal melt rate contribute to $\mathrm{D}H_i/\mathrm{D}t$ and this only shows slight differences, a temporal variation in basal melting can be excluded as the reason for the significant discrepancies that we find. Furthermore, this indicates that the techniques derive consistent changes in ice thickness from their initial measurements after applying the corrections for the surface mass balance and the firn densification, and that the large differences in basal melt rates result principally from differences in the strain rate, which can be improved by the use of modern surface velocity products.

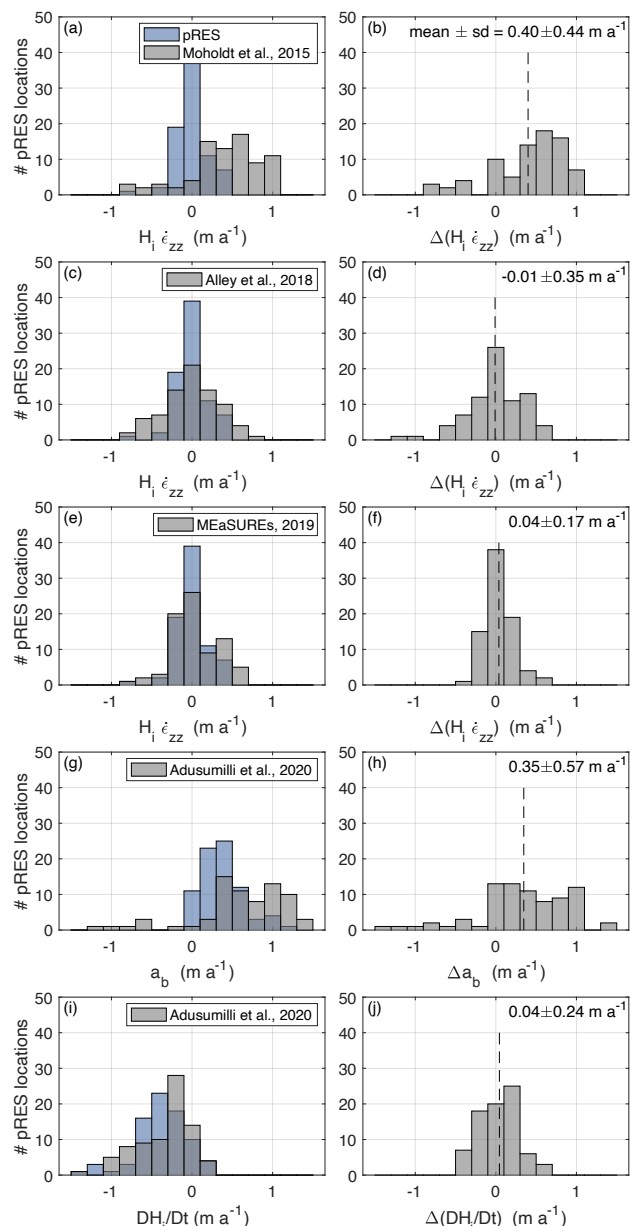

**Figure 5.** Comparison of remote sensing (grey) and pRES-derived (blue) results. The left column shows the distributions (a) of the dynamic change in ice thickness $H_i\dot{\varepsilon}_{zz}$ for the results published by Moholdt et al. (2015), (c) Alley et al. (2018) and (e) derived from the MEaSUREs product (Mouginot et al., 2019a, b), (g) of the basal melt rate $a_b$ and (i) of the change in ice thickness $DH_i/Dt$, both in comparison with those from Adusumilli et al. (2020). The right column (b,d,f,h,j) shows the distribution of the deviation between remote sensing and pRES-derived values according to (a,c,e,g,i). The numbers in the upper right corner state the mean value and the standard deviation (sd). Positive value refer to larger numbers derived from the remote sensing-based method.

## 5 Conclusions

We have presented the first spatial distribution of basal melt rates in the southern Filchner Ice Shelf derived from repeated phase-sensitive radar measurements. In general the melt rates are moderate with maximum values in the centre of less than $1.13\,\mathrm{m\,a^{-1}}$. We tested the representativeness of individual measurements by assessing the variability over short distances. Spatial variability in $a_b$ is low, with occasional outliers possibly linked to large basal gradients. This gives us confidence that a small number of widely spaced measurements accurately represent the large scale melt pattern. Temporal variability, however, is not captured.

Our in situ measurements reveal that inaccuracies in the estimation of dynamic ice thickness change negatively affected recent remote sensing-derived melt rates at our study area at the Filchner Ice Shelf. A comparison with strain rates published by Alley et al. (2018) and with those derived from the newest MEaSUREs velocity field indicates that these inaccuracies can be overcome by using state-to-the-art velocity fields, in which data gaps could be closed. Our study demonstrates that satellite-derived basal melt rates hold great promise, but care needs to be taken, as modelling of the future contribution of Antarctica to sea level rise is currently calibrated using such products (Jourdain et al., 2020). This highlights the need to obtain more data sets such as the one presented here, from across different ice shelves, and to conduct repeated field surveys to assess temporal variability.

*Code and data availability.* MATLAB routines for pRES processing are available from the corresponding author on request. Raw data of the pRES measurements and derived melt rates (https://doi.org/10.1594/PANGAEA.930735) are available at the World Data Center PANGAEA. Echograms recorded with a Center for Remote Sensing of Ice Sheets (CReSIS) Multi-channel Coherent Radar Depth Sounder (MCoRDS) within NASA's Operation IceBridge (OIB) campaign in 2016 can be accessed at https://nsidc.org/data/IRMCR1B/versions/2 (Paden et al., 2014, updated 2019) (last access: 25 April 2021). Basal melt rate data published by Adusumilli et al. (2020) can be accessed at https://doi.org/10.6075/J04Q7SHT (last access: 04 March 2021). Ice-shelf divergence and thickness data published by Moholdt et al. (2015) can be accessed at https://doi.org/10.21334/npolar.2016.cae21585 (last access: 29 April 2021). Strain rate data published by Alley et al. (2018) can be accessed through open ftp by contacting the National Snow and Ice Data Center (NSIDC) (last access: 25 June 2021). MEaSUREs velocity product can be accessed at https://nsidc.org/data/nsidc-0754/versions/1 (Mouginot et al., 2019a) (last access: 13 April 2021). BedMachine Antarctica product can be accessed at http://nsidc.org/data/nsidc-0756 (Morlighem, 2020) (last access: 12 April 2021)

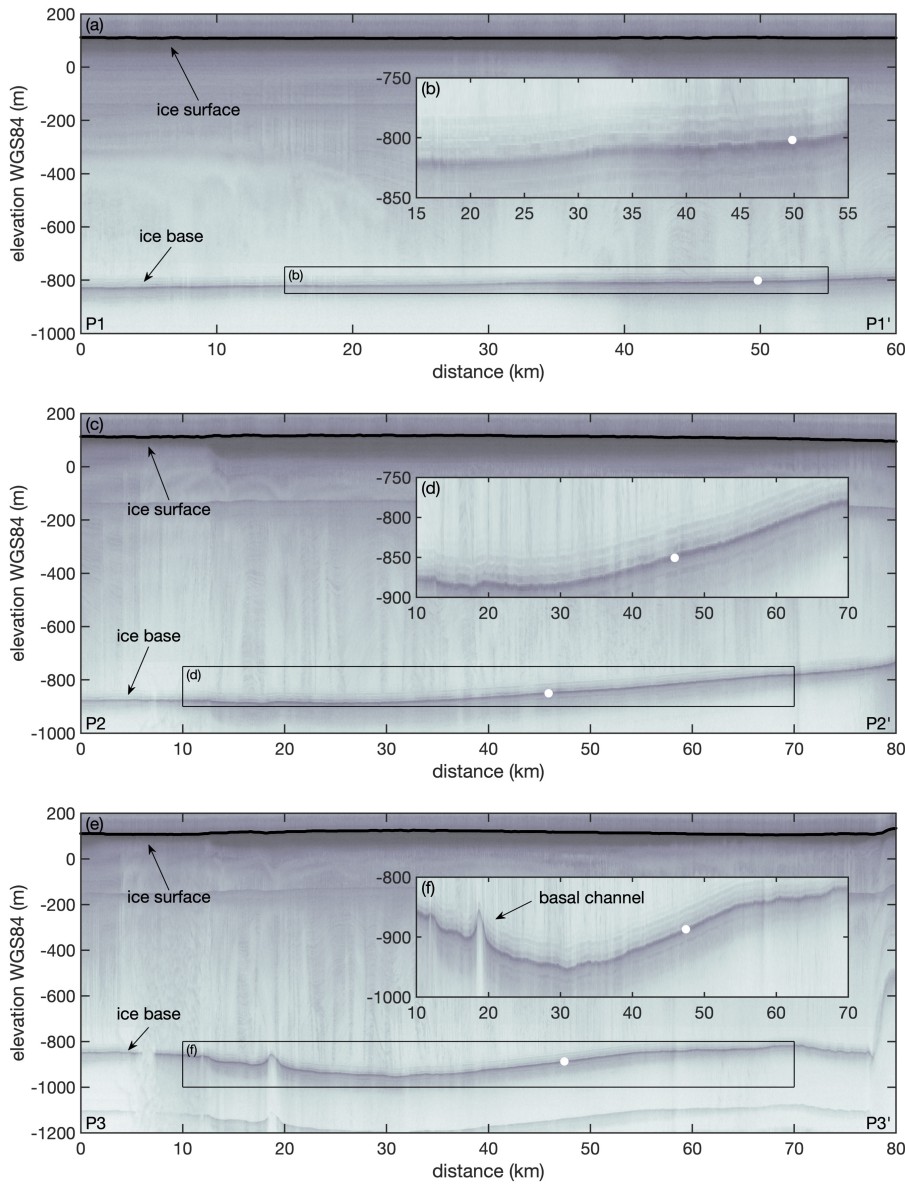

**Figure A1.** Airborne radar echograms (a) P1, (c) P2 and (e) P3 (location in Fig. 1), recorded with a Multi-channel Coherent Radar Depth Sounder (MCoRDS) as part of NASA's Operation IceBridge (OIB) campaign in 2016 (Paden et al., 2014, updated 2019; Arnold et al., 2020). (b,d,f) Insets showing enlarged basal section visualised by black box in (a), (c) and (e). The white dots mark the depth of the ice base derived from a near-by pRES measurement.

## Appendix B:  pRES echograms

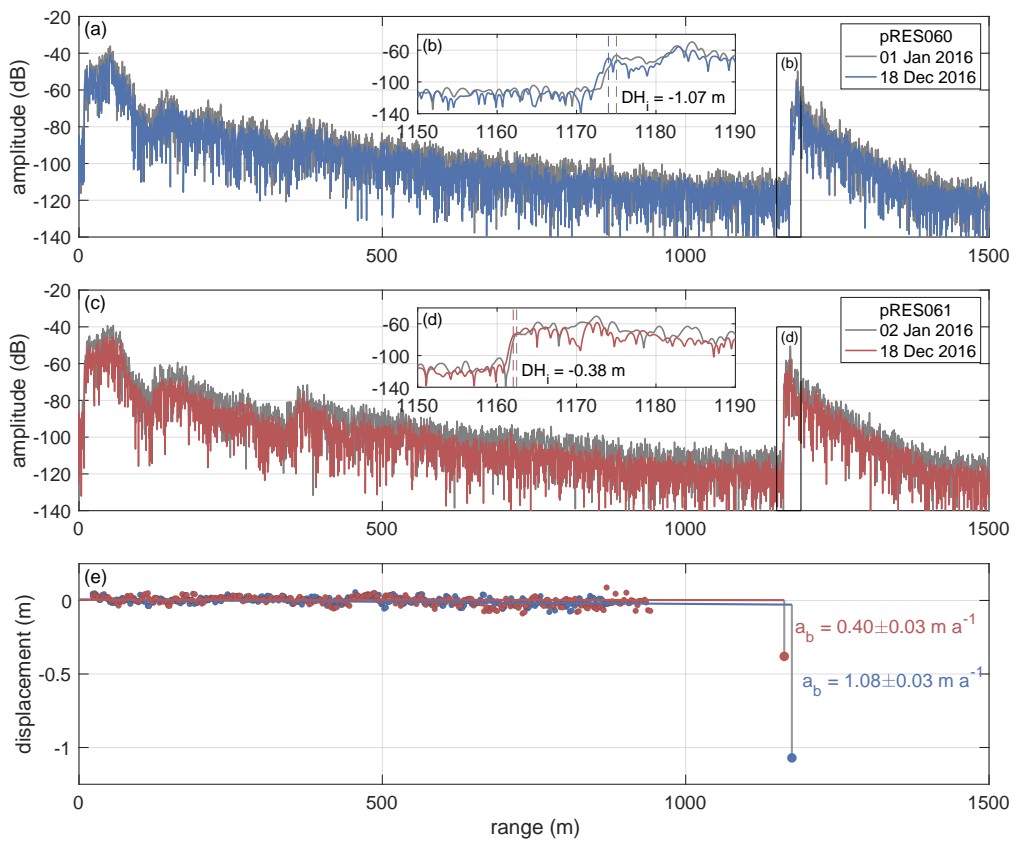

**Figure B1.** (a–d) Amplitude profiles of first (grey line) and repeated measurement at locations pRES060 (a, b; blue) and pRES061 (c, d; red). Insets in (b) and (d) showing enlarged basal section, visualised by black boxes in (a) and (c). (b, d) Vertical dashed lines mark the ice thickness and $\mathrm{D}H_i$ the change in ice thickness between both visits. The correlation coefficients of the basal segments are 0.95 (pRES060) and 0.96 (pRES061). (e) Vertical displacements of internal (small dots) and of the basal segment (large dot) for pRES060 (blue) and pRES061 (red). The values given are the basal melt rates at both locations.

## Appendix C: Comparison with remote sensing

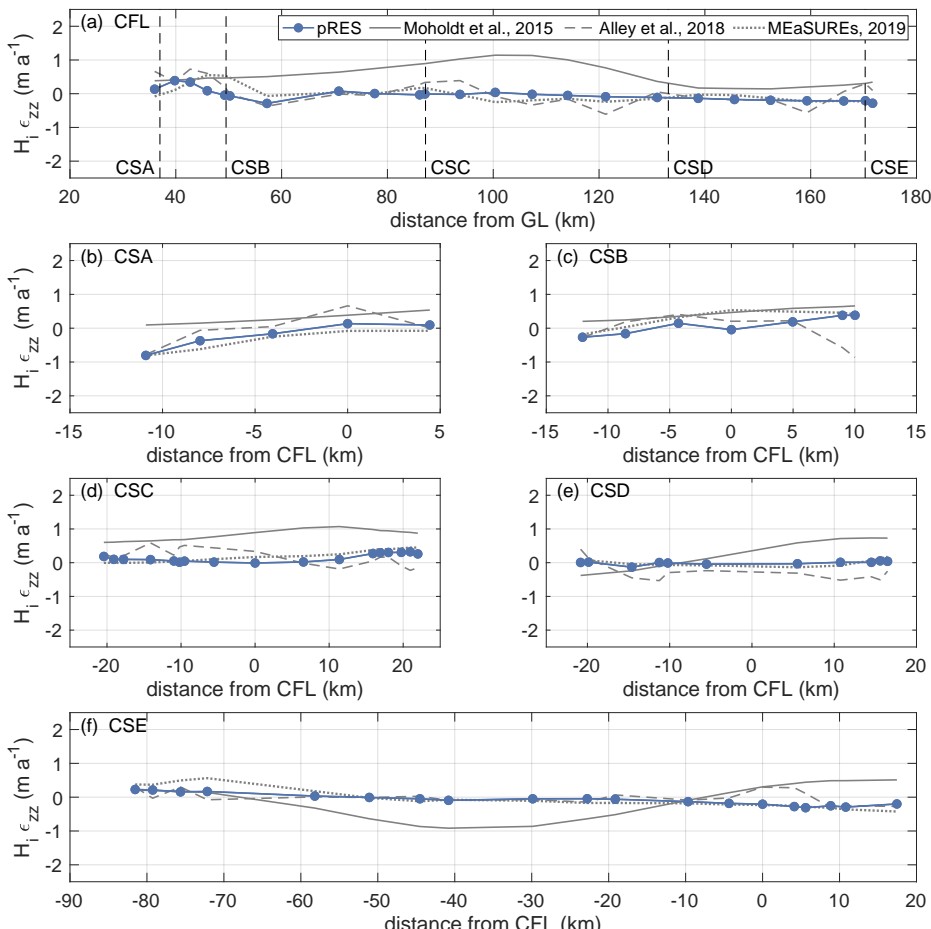

**Figure C1.** Variation of the dynamic ice thickness change $H_i \dot{\varepsilon}_{zz}$ (a) along the Central Flow Line (CFL) of Support Force Glacier's extension on the FIS and (b – f) the cross-sections CSA – CSE. Locations are shown in Fig. 1. pRES-derived values are shown in blue. Remote sensing-derived values are represented by the solid grey line for results published by Moholdt et al. (2015), by a dashed line for results published by Alley et al. (2018), and by a dotted line for estimations derived from the MEaSUREs product (Mouginot et al., 2019a, b). The bounds of the results from Moholdt et al. (2015) display the uncertainties. Derived errors of the pRES measurements are too small to visualise. For CFL, the distance refers to the grounding line (GL) of Support Force Glacier and for all cross-sections to the CFL with positive distances on the eastern side.

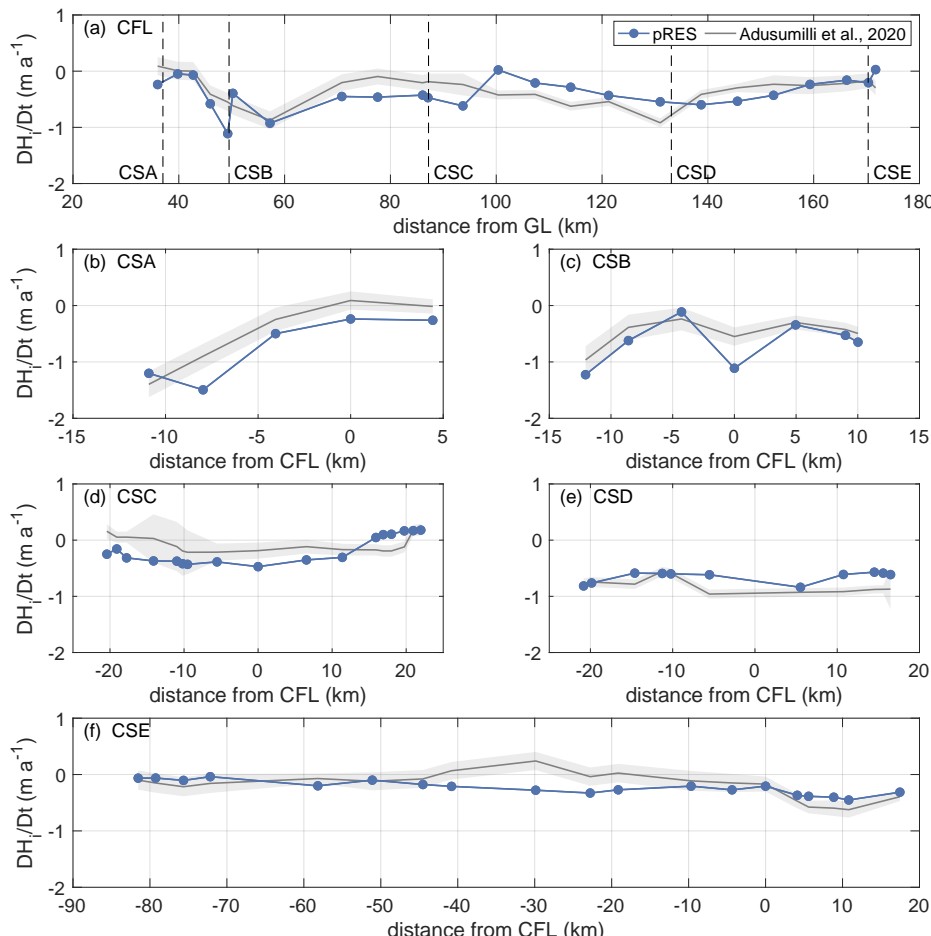

**Figure C2.** Variation of the ice thickness change $\mathrm{D}H_i/\mathrm{D}t$ (a) along the Central Flow Line (CFL) of Support Force Glacier's extension on the FIS and (b – f) the cross-sections CSA – CSE. Locations are shown in Fig. 1. pRES-derived values are shown in blue. Remote sensing-derived values are represented by the dark grey line for results published by Adusumilli et al. (2020). The light grey bounds display the uncertainties. Derived errors of the pRES measurements are too small to visualise. For CFL, the distance refers to the grounding line (GL) of Support Force Glacier and for all cross-sections to the CFL with positive distances on the eastern side.

*Author contributions.* OZ processed the data. OZ and AH analysed the basal melt rates and wrote the manuscript together with KWN. AH has designed the study and conducted the field study together with DS. KWN supported the field study and contributed to melt rate analysis
and its discussion together with HFJC and AH. CLS wrote main parts of the MATLAB routines and supported data analysing. All authors reviewed various versions of the paper.

*Competing interests.* The authors declare that they have no conflict of interest.

*Acknowledgements.* This work was funded by the AWI strategy fund project FISP. Support for this work came from the UK Natural Environment Research Council large grant "Ice shelves in a warming world: Filchner Ice Shelf System" (NE/L013770/1). We acknowledge
provision of data products by Susheel Adusumilli (Scripps Institution of Oceanography, University of California San Diego, La Jolla, CA, USA), that have been published in Adusumilli et al. (2020). We want to thank Graham Niven and Bradley Morrell for their support in the field. We are grateful for discussions with Veit Helm and Niklas Neckel (AWI) on remote sensing-derived vertical strain rates and our results. We thank the two anonymous reviewers for careful reading and very helpful suggestions.

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
