# Peer review of "Basal melt of the southern Filchner Ice Shelf, Antarctica"

_The Cryosphere, 2021_

## Referee Comment (RC1)

**Comments to "Basal melt of the southern Filchner Ice Shelf, Antarctica" by Ole Zeising et al.**

**1  General comments**

In this paper, the authors presented direct pRES measurements of basal melt rates on 79 locations of Filchner Ice Shelf. The study region covered a rather large region between ∼50 km downstream of Support Force Glacier and Berkner Island. Two pRES measurements are conducted in 92 locations repeatedly between about two years. The distribution of basal melt rates suggest in the study region, basal melt rates are quite evenly distributed. Results are also compared to the basal melt rates produced by remote sensing data sets. It's suggested that the quality of basal melt rates derived from remote sensing data sets is limited by the precision of vertical strain rate, and this can be improved by using state-of-art velocity fields.

The data set is a very useful product to constrain ocean model and remote sensing products in this region. However, there are some technical questions need to be addressed, such as the uncertainties of the measurements due to the methodology. Furthermore, ocean mechanism in this region and its impact on the basal melt rates should be better explained. Details will be mentioned in the specific comments.

**2  Specific comments**

L14: 'buttressing of'→ 'buttressing to'

L20: 'satellite remote sensing data'→'remote sensing data'

L45: Can you give more details of the measurements either here or in the method session? For example, how did you track the location of the measurements, using bamboos or gps coordinates...? Shifts of locations might leads to ill alignment of the basal layer.

Just curious, why no measurement is conducted closer to the grounding line of SFG, where more variability may appear? Is it a security consideration?

L65: 'location' → 'locations'

L66: How much does the periods influence melt rates? 323 days are more than a month less than a year. In the previous study of Sun et al., (2019) where hourly measurement of 1-year length is conducted in Roi Baudouin ice shelf, there is a clear seasonal variation in the time series. Melt mainly happens in summer, while in winter time the melt rate is close to zero and refreezing happens. That means without contribution of 42 melting time in summer, melt rates may be underestimated.

L66: 'between 323 and 356' → 'from 323 to 356'

L76 and equation (1): Here vertical strain rate is assumed to be constant within the whole ice column. How good is the assumption hold? Can you show the displacements of the internal layers? Maybe add a subplot in figure B1.

Fig 3: Could the authors add the error bar to the melt rates?

L101, Fig 3: Could the authors add the ice draft measured by pRES e.g. by using two y-axis?

L104-108: It's not clear to me how the vertical gradients of the ice temperature influence the distribution of basal melt rates distribution. And how does it explain the melt rates distribution observed in this article?

L111: What are the locations of the 18 pRES measurements? Can you

show them in the map?

L114: 'and' → 'an'?

L118: For Fig.3, again, it would be straight forward to demonstrate the potential influence of geometry if the authors could add the ice draft of these locations. For Fig. B1, the two signals in the inset plot (d) are not very similar visually. How well is the correlation?

L119: The authors suggest that the higher melt of pRES016 is due to deeper ice draft and therefore higher thermal forcing. I hope the authors can discuss the impact of ocean dynamics to melt rates in this region, including thermal forcing, ocean circulation, and the influence of local boundary (depth of ice draft, slopes).

L120-122: Should this be a localized phenomenon? Will this conclusion hold at locations closer to the grounding line?

Fig.5: With melt rates and vertical strain rate difference between pRES and remote sensing results, can you add another column of the differences caused by surface mass balance? This would make this figure more informative.

**References**:

Sun, Sainan; Hattermann, Tore; Pattyn, Frank; Nicholls, Keith W. ; Drews, Reinhard; Berger, Sophie. 2019 Topographic shelf waves control seasonal melting near Antarctic Ice Shelf grounding lines. Geophysical Research Letters, 46 (16). 9824-9832. https://doi.org/10.1029/2019GL083881

---

## Author Response (AR1)

green: original response to Referee comment
red: changes made in the revised version

**Authors point-to-point response on Referee Comment #1 to tc-2021-230**

1. General comments

In this paper, the authors presented direct pRES measurements of basal melt rates on 79 locations of Filchner Ice Shelf. The study region covered a rather large region between ~50 km downstream of Support Force Glacier and Berkner Island. Two pRES measurements are conducted in 92 locations repeatedly between about two years. The distribution of basal melt rates suggest in the study region, basal melt rates are quite evenly distributed. Results are also compared to the basal melt rates produced by remote sensing data sets. It's suggested that the quality of basal melt rates derived from remote sensing data sets is limited by the precision of vertical strain rate, and this can be improved by using state-of-art velocity fields.

The data set is a very useful product to constrain ocean model and remote sensing products in this region. However, there are some technical questions need to be addressed, such as the uncertainties of the measurements due to the methodology. Furthermore, ocean mechanism in this region and its impact on the basal melt rates should be better explained. Details will be mentioned in the specific comments.

2. Specific comments

L14: 'buttressing of'→ 'buttressing to'
*Thanks! We corrected this.*

L20: 'satellite remote sensing data'→'remote sensing data'
*Thanks! We corrected this.*

L45: Can you give more details of the measurements either here or in the method session? For example, how did you track the location of the measurements, using bamboos or gps coordinates...? Shifts of locations might leads to ill alignment of the basal layer.

*In the revised version, we added more information about the measurement itself.*

*L58:*
*"The pRES transmits a frequency modulated sweep (chirp) from 200 to 400 MHz over a period of one second **via two skeleton slot antennas, separated by roughly 9 m. The exact locations were marked with two bamboos for precise relocation a year later.**"*

Just curious, why no measurement is conducted closer to the grounding line of SFG, where more variability may appear? Is it a security consideration?

> *Indeed, it has been due to a large crevasse field that is south of our first crossing. There is a region further west, where one could potentially conduct such measurements until the grounding line, but this was just figured out by a seismic campaign a year later. If we could redo such a campaign, we would then try to set up an ApRES at those locations. However, that is linked to a melt channel and won't give a 'background' field of melt rates.*

L65: 'location' → 'locations'
> *Thanks! We corrected this.*

L66: How much does the periods influence melt rates? 323 days are more than a month less than a year. In the previous study of Sun et al., (2019) where hourly measurement of 1-year length is conducted in Roi Baudouin ice shelf, there is a clear seasonal variation in the time series. Melt mainly happens in summer, while in winter time the melt rate is close to zero and refreezing happens. That means without contribution of 42 melting time in summer, melt rates may be underestimated.

> *Thanks for raising this point. It is true that our measurements may underestimate the melt rate in case of enhanced melt rates in summer. While there are up to 356 days between the repeat measurements in the southern area, this period is somewhat shorter in the northern part of our study area with 323 days. The period that was not covered by these measurements is between the beginning of December and mid-January. However, due to a lack of time series in the vast part of our survey area, we cannot assess if we indeed underestimate it. In the dataset we are providing, the date of first and second measurement is given and the interested reader has all detailed info.*

> *For the northernmost cross-section (CSE), an autonomous pRES (ApRES) measurement was conducted by our partners of BAS in the same time period as our pRES measurements. This ApRES was located next to a single-repeated pRES measurement that indicates refreezing. However, the derived melt rate from the ApRES time series shows no enhanced melting in summer.*

> *In the revised version, we added a discussion on the possibility of a seasonal cycle that might influence the derived melt rate.*

> *L102 – 111:*
> *This study focuses on the spatial variability of the melt rates, rather than the overall, annual average values, since we did not measure the interannual variability. However, the different sites were occupied for different periods and thus, seasonality in basal melt would affect spatial variability if it exists. The time periods are ranging from 365-9 days to 365-42 days with increasing data acquisition interval southwards. Seasonality may affect the derived annual melt rate differently at the different sites. However, in the northern part of our study area an autonomous pRES (ApRES) station recorded for more than a year, including the period of the single-repeated*

*pRES measurements (Fig. 1b, Vankova et al., 2020). For the time period between 18 Jan 2016 and 06 Dec 2016 (same period than the pRES measurements with the shortest time interval), we derived a melt rate of 0.02±0.03 m/a. In the period of 365 days (18 Jan 2016 -- 17 Jan 2017), a slightly lower melt rate of 0.01±0.03 m/a was derived. This indicates that no enhanced melting occurred at the location of the ApRES in summer 2016/2017. However, we cannot assess if melt rates are enhanced at other locations.*

L66: 'between 323 and 356' → 'from 323 to 356'
 *Thanks! We corrected this.*

L76 and equation (1): Here vertical strain rate is assumed to be constant within the whole ice column. How good is the assumption hold? Can you show the displacements of the internal layers? Maybe add a subplot in figure B1.

 *Thanks for raising this point. In the analysis of the data, we carefully checked if a constant strain is valid for all stations. At no station, the displacement distribution of the internal layers differ significantly from the fitted line.*
 *We added the vertical displacements of the stations pRES060 and pRES061 to another subfigure in figure B1. Since the vertical strain rate is low at this location as well as at many other stations, the displacements are close to zero.*

Fig 3: Could the authors add the error bar to the melt rates?

 *There are error bars but the error is too small to see as the dot marking the pRES derived melt rate is of similar size than the error bar.*
 *We updated the sentence in the caption as follows:*
 *"Uncertainties of the pRES derived melt rates are 0.03 cm and therefore too small to visualise."*

L101, Fig 3: Could the authors add the ice draft measured by pRES e.g. by using two y-axis?
 *Thanks! Yes, we added the ice draft to Fig. 3.*

L104-108: It's not clear to me how the vertical gradients of the ice temperature influence the distribution of basal melt rates distribution. And how does it explain the melt rates distribution observed in this article?

 *The basal melt rate is determined by the energy balance at the ice-ocean interface. One term in this energy balance is the vertical temperature gradient, which is the heat flux at the ice side. The larger the temperature gradient, the lower the basal melting rate, as more energy is used to heat the ice.*
 *Depending on the magnitude of the oceanic heat flux, the influence of the gradient of the ice temperature may be insignificant, or may contribute significantly. Ice streams often exhibit a cold core and this temperature distribution is altered over time and hence distance from the grounding line in flow direction, as also shown by Humbert*

*(2010). Over distance, the temperature gradient in the ice is reduced as the temperature profile in case of basal melting is approaching a parabolic shape. With decreasing vertical gradients of the temperature in the ice, its influence is declining and less energy is needed to heat the ice. This favours higher melt rates.*
*In order to assess the influence in our study area, one would need vertical ice temperature profiles, which have not been measured as this requires drilling.*

*In the revised version we added a discussion on the different mechanisms that affect the basal melt variation.*

*L126 – 143:*
*In an ice shelf cavity where the water speeds are relatively high as a result either of tides, as in this case (Mueller et al., 2018), or as a result of strong buoyancy-driven flows, as in the case of a steeply-inclined ice base over relatively warm water (Lazeroms et al., 2019), basal melt rates are mainly controlled by three factors: the basal drag coefficient, the thermal driving, and the water speed in the boundary layer. The thermal driving is the difference in the temperature of the water near the ice base and the freezing point of that water at the pressure of the ice base. The water speed and the basal drag generate the shear-driven turbulence that efficiently diffuses heat and salt towards the ice base (Holland and Jenkins, 1999). A fourth factor, discussed below, is the basal vertical temperature gradient in the ice.*

*In our study area the basal slopes are generally low (Morlighem 2020; Morlighem et al., 2020), as is the thermal driving. We therefore expect tidal speeds to dominate buoyancy-driven flows. Ice draft plays the role of modifying the thermal driving: lower basal pressures reduce the freezing point, thereby reducing the local freezing temperature. Mueller et al. (2018) find a strong increase in tidal speeds from the grounding line to the Cross-Section CSE; this parallels a reducing basal draft, which will act to reduce the thermal driving. We expect these two tendencies to work together to modulate the large scale spatial variation in basal melt rates.*

*The large scale spatial variation in a_b can also be influenced by changes in vertical gradients of the ice temperature. An ice shelf fed by a fast glacier typically contains a cold core as a result of ice advection, leading to larger vertical temperature gradients some distance from the grounding line. However, with melting over centuries, the ice temperature is more likely to approach a parabolic profile, with only moderate temperature gradients (Humbert, 2010).*

L111: What are the locations of the 18 pRES measurements? Can you show them in the map?

> *Yes, we highlighted the nearby station pairs in the map. Thanks!*

L114: 'and' → 'an'?

> *Thanks! We corrected this.*

L118: For Fig.3, again, it would be straight forward to demonstrate the potential influence of geometry if the authors could add the ice draft of these locations.

*We added the ice draft to Fig. 3, however, the change in ice draft of the pRES measurements shown in Fig. B1 is hardly visible, since the difference is small compared to the change over the entire central flow line.*

For Fig. B1, the two signals in the inset plot (d) are not very similar visually. How well is the correlation?

*The correlation is calculated for the basal segment ranging from -9 to +1 meter of the basal reflection (first maximum in amplitude after the strong increase). The correlation coefficient of the basal segment of pRES061 is 0.95. The correlation coefficient of the nearby station pRES060 is 0.96 and thus only slightly higher.*

*We added these numbers in the caption of Fig. B1.*

L119: The authors suggest that the higher melt of pRES061 is due to deeper ice draft and therefore higher thermal forcing. I hope the authors can discuss the impact of ocean dynamics to melt rates in this region, including thermal forcing, ocean circulation, and the influence of local boundary (depth of ice draft, slopes).

*Thanks for raising this point. In the revised version, we added a discussion on the contribution of the different oceanographic and glaciological mechanisms to the small scale spatial variability.*

*L156 – 164:*
*Variability at small spatial scales will not result from variations in tidal speed: in the absence of strong sea floor or ice base topography, strong horizontal gradients in tidal speed are not expected. As previously discussed, tides will dominate buoyancy-driven currents, and are therefore also unlikely to play a significant role in controlling local variations in melt rate. However, through its effect on thermal driving even a quite modest local variation in basal depth is a candidate for driving variation in basal melting. A change in ice draft of, say, 10 m metres will change the thermal driving by about 0.007 °C (e.g. Holland and Jenkins, 1999). Using the algorithm proposed by Jenkins et al. (2010) for the nearby Ronne Ice Shelf, and for an appropriate mean tidal speed of 0.1 m/s (Mueller et al., 2018), a 0.007 °C change in thermal driving would result in a melt rate difference of 0.17 m/a.*
*A second candidate driver of local variability in melt rate is spatial variability in basal roughness. Differences in the drag coefficient at the ice base will directly affect melt rates (e.g. Holland and Jenkins, 1999).*

L120-122: Should this be a localized phenomenon? Will this conclusion hold at locations closer to the grounding line?

*We have only discrete measurements and our measurements are conducted at quite some distance to the grounding line due to a massive crevasse field further south. If*

*a basal crevasse is the origin of the localised change in ice thickness, bending in the hinge zone has likely led frequently to formation of crevasses and it happens that we just by chance covered one of such features in our measurements. If this is the case or not can only be assessed by (a) having a continuous radargram from the grounding line northwards retrieved from an airborne campaign and (b) some more measurements in the vicinity of basal crevasses and (c) a more continuous profile of pRES measurements or (d) ApRES measurements spaced in 2[a]\*v[m/a] distance, running over two years.*

Fig.5: With melt rates and vertical strain rate difference between pRES and remote sensing results, can you add another column of the differences caused by surface mass balance? This would make this figure more informative.

*Thanks for raising this point. Unfortunately, the comparison of the surface mass balance is not possible as a separation between surface mass balance and densification is not possible from the pRES measurement, as only the sum of both is derived. However, the comparison of the pRES- and remote sensing-derived melt rates does not require this comparison. The starting point of the comparison is a product derived from the change in ice thickness and the correction of the surface mass balance and the snow and firn densification. Thus, this product represents the change in ice thickness in the ice column and it is shown in Fig. 5i and Fig. C2.*

References:

Vaňková, I., Nicholls, K. W., Corr, H. F., Makinson, K., & Brennan, P. V. (2020). Observations of tidal melt and vertical strain at the Filchner-Ronne Ice Shelf, Antarctica. *Journal of Geophysical Research: Earth Surface*, *125*(1), e2019JF005280, https://doi.org/10.1029/2019JF005280.

**Authors point-to-point response on Referee Comment #2 to tc-2021-230**

In this paper, Zeising et al. present the first ground-based determination of basal melt rates of the southern Filchner Ice Shelf using repeat phase-sensitive radar measurements. They find low spatial variability in melt rates, with net freezing at only three closely spaced sites. They compare their calculated melt rates to those determined by satellite remote sensing and find that the discrepancies can mostly be explained by errors in the velocity field used in the other studies. Thus, they conclude that (1) basal melt rates determined at a single location are likely a good indication of large-scale melt rates, and (2) melt rates determined by satellite remote sensing should use the latest velocity datasets to improve accuracy.

This paper is valuable, well written, and uses novel methods. I recommend publication in The Cryosphere after some minor revisions. I have two main concerns that should be addressed. The rest of my comments are mostly line edits or requests for some clarification of details.

First, I don't fully understand the analysis presented in Section 3.2 and Figure 4. This could be because there doesn't seem to be a trend in the data (perhaps simply because of the scale of the vertical axis), and the plot is parametric with two other quantities. My confusion about the plot is exacerbated by the somewhat confusing text in this section. I think what the reader is supposed to understand is that the values are small and there are no discernable trends, but the presentation of the data made it difficult for me to arrive at this interpretation. The authors should consider revising this plot (perhaps with multiple panels to help the reader, rather than putting all of this in one plot) and making the text of this section clearer. There may also be a more suitable choice of independent variable than difference in draft between locations. I go into more specifics in my detailed comments, below.

*Thanks for raising this point. We will address it in the specific comments below.*

Second, the Discussion section (Section 4) of the paper is limited to comparing the inferred melt rates with those determined by satellite remote sensing. This is a very useful comparison and leads to practical conclusions; however, I feel that there should be some more discussion of what the melt rates indicate about the oceanographic and glaciological conditions. There is very little context given for these melt rates and the amount of variability.

What do the results (especially the low spatial variability and lack of higher melt rates close to the grounding line) tell us about melt-rate parameterizations used by numerical models, like those used by ISMIP6 (Jourdain et al., 2020) or the plume-based parameterization of Hoffman et al. (2019)?

*First of all we would like to point out that we cannot rule our higher melt rates close to the grounding line – our measurements are still quite a bit away from the grounding line. The parameterisation in numerical models has been calibrated against remote sensing melt rates and we discuss in this manuscript how those fit to our in-situ observations. Therefore, we actually discuss how good the data basis was for the derivation of the parameterisation. If a particular model in ISMIP6 was forced with a good or poor basal melt rate distribution depends, however, also on how well the model represented the ice thickness in the area and not only how good the parameterisation was. Even a perfect parameterisation would lead to poor forcing if the ice thickness in the model is over- or underestimated. All that is not the topic of*

*this manuscript. We think that our data, which is freely available, will help modellers to cross-check how much off their forcing is/was and will be a good data basis in this area for revised versions of the parameterisation.*

I think the impact of this paper would be increased by adding some discussion of how these melt rates relate to physical processes and our understanding of and ability to model this system.

*We do understand that this is desirable and we would very much like to achieve this. But with not having measurements from e.g. moorings in the ocean or the ice temperature in that area, this remains speculative. However, we added a discussion of the different physical processes influencing the (small scale) spatial distribution of the basal melt rats (see comments below).*

*Nevertheless, the measurements can now be used to conduct simulations, like for example stand alone ice sheet simulations and the resulting ice temperature distribution can be analysed to assess gradients in the ice. Still, this may be highly influenced from lack of knowledge of geothermal heat flux on the inland ice side, but it is something that could be investigated in the future. In addition, ocean models can use this distribution of melt rates also as a benchmark experiment and can investigate which mechanisms need to be large or small to obtain this spatial distribution in melt rates. This type of study is very different from what we present here and needs to be done by different teams, but we look very much forward to such studies.*

Detailed comments:

Line 42: Unusual use of "benign". What is meant by this?
*We used "benign" to express that the southern part of the Filchner part is better accessible, as crevasse fields prevented a survey with ski-doos in the northern part of the ice shelf. We have rephrased this to 'accessible'.*

L 67: What is meant by "low correlation chirps"?
*With "low correlation chirps" we mean chirps that have a low correlation coefficient on average when the chirp is correlated with every other chirp. As these chirps cause noise in the amplitude- and phase profiles, we removed them before stacking.*

*We admit that this can be better expressed and we changed the sentence to:*
*L69:*
*"Correlations were calculated between each chirp and the 99 remaining chirps, and those chirps with a low average correlation were discarded."*

L 70: Do you have an estimate of the uncertainty in your range estimate based on the Herron and Langway model?

*An assessment of the uncertainty caused by the use of a density model is not easily possible. As a result of higher propagation velocities of the electromagnetic wave in the firn, the ice thickness is reduced by a few meters compared to a constant propagation velocity that is adapted to the density of ice. Using a density model, from which the propagation velocities are derived, ensures a somewhat more realistic ice thickness. However, the uncertainty of the propagation velocity is still around 1%.*

*Nevertheless, the uncertainty of the ice thickness has only a minor influence on the determination of the melt rate, since the change in the ice thickness is independent of this. The influence is limited to the dynamic ice thickness change, since the vertical strain is multiplied by the ice thickness.*

*We added the following sentence to the manuscript:*
*L73:*
*"Nevertheless, the uncertainty of the propagation velocity is about 1% (Fujita et al., 2000)."*

L 80: "plain strain" should be "plane strain"
*Thanks! We corrected this.*

L 85–86: Add citation for why this is reliable for plug flow.
*That is a good idea. We are referring in the revised version to Greve & Blatter.*

L ~90: Would it be possible to use the measurements from the 15 excluded stations to calculate minimum and/or maximum melt rates at these locations?

*For some stations, an estimation of a minimum and maximum melt rate would be possible. Especially at those ~5 locations at which low correlation might have led to a half-wavelength ambiguity due to phase wrapping. However, the error would increase by 0.28 m in order to take this uncertainty into account. Since we have observed low melt rates in general, this would be a significant proportion and thus the station would have been discarded for the comparison with remote sensing data. At three other stations where the vertical strain could not have been determined, vertical strain rates from nearby stations could have been used. However, this also leads to uncertainties and an exclusion from the comparison. At all of the other excluded stations, it was not possible to determine the alignment and the strain or of the change in ice thickness, and thus no estimate of the melting rate either.*

What are the possible physical explanations for reasons (1) and (3)?

*Explanations for the low correlation values that were the cause of criteria (1) and (3) could be errors in operating the pRES, such as inaccuracies in the alignment of the*

*antennas or incorrectly seated cables. Changes of the settings such as the attenuation are also conceivable as a cause.*

*There is no study considering this so far, but one can conceive the following effect to be responsible for the issues around the pore-close off, which is the firn-ice transition. When pores are closed off, the scattering mechanism is changing from cylindrical scatterers (pores) to spherical scatterers. One can imagine that this also affects the amplitude of the retrieved signal in that depth. This way, around the pore-close off, the amplitude is changing, which causes these issues, while further up and down in the firn/ice column, the scattering mechanism is not changing. This is to our knowledge not yet been discussed in literature, but in simulation of scattering in satellite geometry and for satellite sensors, is that taking into account by mixing the scatterer types. Unfortunately, this is not directly comparable to our situation.*

*Possible reasons within the ice that might have caused low correlation value are for example, strong deformation or shear. However, regarding the condition at the Filchner Ice Shelf and the good correlation values found at nearby stations, we think that this can be ruled out.*

Fig 3 caption: Would be helpful to give the order of magnitude or the range of aPRES uncertainty

*Thanks for raising this point. We updated the sentence in the caption as follows: "Uncertainties of the pRES derived melt rates are 0.03 cm and therefore too small to visualise."*

L 103: I only see two "freezing" datapoints in Fig 3. Is this just because two of those stations are very close together (Fig 1)?

*Seven locations were chosen to be nearby another station for the analysis of the small scale spatial variability, but slightly outside the cross-section or the central flow line. Therefore, we initially left these out of Fig. 3. As this obviously leads to irritation, we have included them in Fig. 3 again.*

L104–108: Is the implication here that the ice temperature gradients are counteracting the expected variation due to ice draft? Can you make the connection between your results and these last few sentences more explicit?

*Thanks for raising this point. The ice temperature gradient is one term in the energy balance of the interface between ice and ocean. As higher this term is, as lower is the basal melt rate. It may, however, be much smaller than the oceanic heat flux and not contribute significantly if the oceanic heat flux is large. In case the oceanic heat flux is small, the term may become more important. A larger draft favours higher melt rates as it reduces the pressure melting point and thus it counteracts the variation caused by the temperature gradient. Nevertheless, this comparison is not trivial since the variation of draft may change ocean dynamics and as a consequence of that the oceanic heat flux into the ice can vary. But as ocean thermodynamics has many*

*components, it is not possible to directly compare the effect of the ice temperature gradient to the effect of the basal topography gradient.*

*In the revised version we added a discussion on the different mechanisms that affect the basal melt variation.*

*L126 – 143:*
*In an ice shelf cavity where the water speeds are relatively high as a result either of tides, as in this case (Mueller et al., 2018), or as a result of strong buoyancy-driven flows, as in the case of a steeply-inclined ice base over relatively warm water (Lazeroms et al., 2019), basal melt rates are mainly controlled by three factors: the basal drag coefficient, the thermal driving, and the water speed in the boundary layer. The thermal driving is the difference in the temperature of the water near the ice base and the freezing point of that water at the pressure of the ice base. The water speed and the basal drag generate the shear-driven turbulence that efficiently diffuses heat and salt towards the ice base (Holland and Jenkins, 1999). A fourth factor, discussed below, is the basal vertical temperature gradient in the ice.*

*In our study area the basal slopes are generally low (Morlighem 2020; Morlighem et al., 2020), as is the thermal driving. We therefore expect tidal speeds to dominate buoyancy-driven flows. Ice draft plays the role of modifying the thermal driving: lower basal pressures reduce the freezing point, thereby reducing the local freezing temperature. Mueller et al. (2018) find a strong increase in tidal speeds from the grounding line to the Cross-Section CSE; this parallels a reducing basal draft, which will act to reduce the thermal driving. We expect these two tendencies to work together to modulate the large scale spatial variation in basal melt rates.*

*The large scale spatial variation in $a\_b$ can also be influenced by changes in vertical gradients of the ice temperature. An ice shelf fed by a fast glacier typically contains a cold core as a result of ice advection, leading to larger vertical temperature gradients some distance from the grounding line. However, with melting over centuries, the ice temperature is more likely to approach a parabolic profile, with only moderate temperature gradients (Humbert, 2010).*

L 112–113: Presumably BedMachine surface elevation and thickness, not surface elevation alone?

*In order to calculate the draft, we used the surface elevation from BedMachine and the pRES-derived ice thickness. We agree that this can be written more clearly.*
*We updated the sentence in the revised version:*

*L148:*
*The draft was derived from the BedMachine surface elevation (Morlighem, 2020; Morlighem et al., 2020) and pRES ice thickness.*

L 115: Both ΔH and Δh are used in this paragraph, and seem to indicate the same quantity.
*Yes, thank you very much for finding this mistake. You are right, ΔH should be Δh.*
We corrected this.

L 113–114 and Figure 4: I don't understand what is meant by Delta h_b indicating "large scale basal topography for the two locations." Is this supposed to give an indication of the overall slope or roughness, or is the change in draft really the variable of interest? For a rough ice base, you could have a Delta h_b of zero between two points even if there was an overall slope that could drive differences in melting. Is this statement contingent on having a smooth ice base (which the CReSIS data indicate is probably the case)? It seems like either a roughness metric, the mean draft, or the mean slope of the ice shelf base would give a better indication of the large scale basal topography.

*You are right, 'large scale basal topography' is misleading when considering on purpose only nearby stations and hence small scale variability. We change the wording in the revised version to 'change in basal topography' over the two locations. In general one has always the issue that roughness on one scale is the topography on another scale. Here we meant to compare topography changes steering water masses, rather than roughness that may contribute to frictional heat. If we would have CReSIS profiles everywhere, we could indeed compute a roughness parameter, but we unfortunately do not have any in the respective areas. Please also see the answer below.*

L 119: What is "Beside" is supposed to indicate here?
*"Beside" was not the right wording here.* Therefore, we removed it from this sentence.

L 121: "many ice thicknesses" might be an overstatement. Based on ice thicknesses in Fig 1, your measurement separations are on the order of 1–3 ice thicknesses.
*We agree, 'many' is just wrong here. We corrected it in the revised version:*

L165:
Overall, this gives evidence that individual measurements are representative of a large area on the scale of 1--3 ice thicknesses.

L: 125: "ice shelf base base"
*Thanks! We corrected this.*

Fig 4. If the outlier around (11, 0.75) is removed, is there a distinguishable trend here? I'm not sure I totally understand this choice of analysis or what I am supposed to understand from how the data are presented. I think I'm supposed to understand that nearby stations have the same thermal forcing, so this is trying to remove that variability to get at the influence from draft alone, which turns out to be small. However, it seems like the values of âhb here are small enough that I wouldn't expect draft to be at all important in explaining the difference in melt rates between sites. I would expect that local oceanographic properties,

ice temperature, local ice base slope, or ice base roughness would be more important than a few-meter change in draft. Of course, those quantities are not readily available from existing data, and so it is difficult to determine a relevant independent variable. Would some more meaningful pattern emerge if you plotted melt rate differences as a function of horizontal distance between sites? Then you could include locations outside of just the stations within 2 km of each other. Or alternatively, you could calculate the ice base slope over O(100m) length scales using the BedMachine draft. Or perhaps the text in Section 3.2 just needs to be revised to explain this figure more clearly.

*Many thanks for your detailed feedback and suggestions for the Section 3.2 and Fig. 4.*

*In this section we want to show the small scale variability of the basal melt rate in order to evaluate the reliability of the large-scale variability. The lower the small scale variability, the more reliable a derived melt rate is for its environment.*

*In case of a large variability, as it is the case at one location, it is important to classify this. As you said correctly, there are several possible reasons: oceanographic properties, ice temperature, local ice base slope, ice base roughness or draft. Due to a lack of data, except for slope and draft, these reasons cannot be further investigated.*

*Therefore we plotted the deviation of the melt rate against the change in the draft - which is the local ice base slope. This analysis shows that the two nearby stations with the largest difference in the basal melt rate are also those with a large deviation in the draft.*

*The aim of this figure was therefore not to show a trend between the difference in draft and the difference in melt rate. It was all about showing that the variability is generally small and that deviations are connected with changes in the draft.*

*Our intention was to plot the change in melt rate over a quantity that might be of interest and that is accessible. The temperature gradient of the base is unknown. Oceanographic quantities, too, so is a roughness not available. Therefore we have selected the draft here.*

*A change in the draft can affect the melt rate for several reasons:*
*(1) The difference in the draft leads to a change in the pressure melting point.*
*(2) Due to the deviation in draft, rising melt water can accumulate at a location with a lower draft and favour low melt rates.*
*(3) A significant change in the draft results in a "steep" slope, although the reverse is not true, as you correctly described. A steep slope favors higher melt rates due to rising currents. However, it is not the melt rate that is high, but the difference in the melt rate between two nearby locations. If this is relevant, then the slope would have to change significantly between the stations. However, the resolution of the BedMachine data is not sufficient to investigate this.*

*In addition, a local deviation in the draft can also be evidence – and not only the reason – of a local variability in melt rate.*

*We addressed this point and added a discussion on the contribution of the different processes to the melt rate.*

*L156 – 164:*
*Variability at small spatial scales will not result from variations in tidal speed: in the absence of strong sea floor or ice base topography, strong horizontal gradients in tidal speed are not expected. As previously discussed, tides will dominate buoyancy-driven currents, and are therefore also unlikely to play a significant role in controlling local variations in melt rate. However, through its effect on thermal driving even a quite modest local variation in basal depth is a candidate for driving variation in basal melting. A change in ice draft of, say, 10 m metres will change the thermal driving by about 0.007 °C (e.g. Holland and Jenkins, 1999). Using the algorithm proposed by Jenkins et al. (2010) for the nearby Ronne Ice Shelf, and for an appropriate mean tidal speed of 0.1 m/s (Mueller et al., 2018), a 0.007 °C change in thermal driving would result in a melt rate difference of 0.17 m/a.*
*A second candidate driver of local variability in melt rate is spatial variability in basal roughness. Differences in the drag coefficient at the ice base will directly affect melt rates (e.g. Holland and Jenkins, 1999).*

*We also improved the text in order to explain the improved figure more clearly.*

L 177: Can you discuss why you cannot extract a rate of freeze-on? Presumably salty ice with high conductivity and/or low density does not allow for determination of the ice base, but it would be helpful to be more explicit about this. The use of "as yet" suggests that there may be some way around these difficulties. Can you elucidate what these might be?

*Thanks for raising this point. We are happy to give more information about how freeze-on influences the radar signal. However, we prefer to do this in the results section instead of in the conclusion.*

*The freeze-on reduces the contrast in dielectric permittivity at the ice base, which influences the amplitude of the basal return. Thus, at those stations at which no melt was observed and the amplitude at the base was reduced, we assume to observe accretion. However, from an ApRES (autonomous pRES) time series, the temporal change of basal amplitude can be investigated (Vankova et al. 2021).*

*In the revised version we addressed this discussion.*

*L120:*
*A key assumption made during pRES processing is that the phase-shift on reflection at the ice-ocean interface remains constant. Although this is valid for a fresh ice/seawater interface, it is a poor assumption for the interface between fresh ice and possibly slushy sea ice, itself underlain by seawater. This means that if either of a pair of measurements is made during a period of freeze-on it is not possible to distinguish the change in the phase of the basal reflection that results from a change in its range, from the phase shift that results from the change in the nature of the*

*basal interface. Thus, we can not determine the amount of accretion at the three sites.*

References:

Hoffman, M. J., Asay-Davis, X., Price, S. F., Fyke, J., and Perego, M.: Effect of Subshelf Melt Variability on Sea Level Rise Contribution From Thwaites Glacier, Antarctica, Journal of Geophysical Research: Earth Surface, n/a, https://doi.org/10.1029/2019JF005155, 2019.

Jourdain, N. C., Asay-Davis, X., Hattermann, T., Straneo, F., Seroussi, H., Little, C. M., and Nowicki, S.: A protocol for calculating basal melt rates in the ISMIP6 Antarctic ice sheet projections, 14, 3111–3134, https://doi.org/10.5194/tc-14-3111-2020, 2020.

Vaňková, I., Nicholls, K. W., & Corr, H. F. J. (2021). The Nature of Ice Intermittently Accreted at the Base of Ronne Ice Shelf, Antarctica, Assessed Using Phase-Sensitive Radar. *Journal of Geophysical Research: Oceans*, *126*(10), e2021JC017290, https://doi.org/10.1029/2021JC017290.

---

## Referee Report (RR1)

I thank the authors for their careful revisions based on the first round of reviews. I still have a few suggestions that should be addressed before publication. These are still mostly to do with Section 3.2, as well as a few other minor comments. Line numbers below refer to the version with tracked changes.

L 95: In my previous review I asked: "What are the possible physical explanations for reasons (1) and (3)?" This question was answered in the authors' response, but please add a brief explanation of these to the text.

L 105: "The time periods are ranging from 365-9 days to 365-42 days with". This is confusing. Should this be read as "365 minus 9" or "365 to 369", or something else?

L 109: should be "same period *as*" rather than "same period *than*"

L124: Should be "marine ice" instead of "sea ice"

Section 3.2 is greatly improved. Thank you. I still have some minor suggestions for presenting this section and Figure 4, but I think they can be easily addressed. Below, text from the Authors' Response is in green:

*In this section we want to show the small scale variability of the basal melt rate in order to evaluate the reliability of the large-scale variability. The lower the small scale variability, the more reliable a derived melt rate is for its environment.*

In this case, it seems like standard deviation of melt rate as a function of radius from a point would be a better way to judge the reliability of a point measurement. This would implicitly include the effect of $\Delta h_b$, without suggesting that $\Delta h_b$ is the primary driver of $\Delta a_b$. However, I am fine with Fig 4 being presented as-is or with some minor changes, with some more explanation in the text (see below)

*Therefore we plotted the deviation of the melt rate against the change in the draft - which is the local ice base slope. This analysis shows that the two nearby stations with the largest difference in the basal melt rate are also those with a large deviation in the draft.*

*The aim of this figure was therefore not to show a trend between the difference in draft and the difference in melt rate. It was all about showing that the variability is generally small and that deviations are connected with changes in the draft.*

However, the data point with the highest draft difference is not obviously a major increase in $\Delta a_b$ relative several of those with $\Delta h_b < 10$m, especially when these uncertainties are taken into account, which makes the effect of draft on melt rate difference unconvincing here. Please present some statistical metric that shows that the $\Delta a_b$ at $\Delta h_b = 14$ m is indeed a significant outlier from the values with $|\Delta h_b| < 10$m.

Also, $\Delta h$ here takes both positive and negative values. A change of draft of -10m would be expected to have an equal and opposite effect as a change of +10m, but the data point near -10m

displays $\Delta a$ of about 0m. Is there a mechanism to explain a step-change in $\Delta a_b$ at $|\Delta h_b| = 10m$? Perhaps one axis (or both) needs to be a fractional change instead of absolute change?

*The aim of this figure was therefore not to show a trend between the difference in draft and the difference in melt rate. It was all about showing that the variability is generally small and that deviations are connected with changes in the draft.*

*Our intention was to plot the change in melt rate over a quantity that might be of interest and that is accessible. The temperature gradient of the base is unknown. Oceanographic quantities, too, so is a roughness not available. Therefore we have selected the draft here.*

Some text should be added in 3.2 that explains that the choice of $\Delta h_b$ as the independent variable in Fig. 4 is largely made from necessity. Otherwise, the reader assumes that $\Delta h_b$ is presented as the controlling factor on melt-rate, which the sudden increased in $\Delta a_b$ at $|\Delta h_b| = 10m$ suggests is not really the case. This sudden increase in $\Delta a_b$ is interesting, but it is rather hard to fathom why $|\Delta h_b| = 10m$ would act as a threshold value, and no mechanism is provided for it in the text. Again, the authors do not claim that it is indeed a threshold value, but the choice of $\Delta h_b$ as the independent variable makes it seem this way to the reader.

It also looks like a similar pattern would emerge if the independent variable was $a_b$ instead of $\Delta h_b$. Could it be that higher melt rates ($> 1m/yr$) also exhibit larger absolute variation (unclear if this is a larger relative variation) because they are driven by processes different from the very low ambient melt rates? It would be helpful to address this possibility in the text.

L 160: This switches from discussing $\Delta a_b$ to $a_b$, which is not the quantity of interest here.

L 165–173: It would help to add a sentence to the end of this paragraph stating that the ice roughness and basal drag coefficient are not known well enough to identify what is in fact driving the variation in melt rates above $\Delta h_b = 10$ m.

L 184: It would be helpful to define what counts as "low" variability, as 0.2 m/yr is larger than the value of $\Delta a_b$ at $|\Delta h_b| = 14m$ in Fig 4 that is referred to as a "major deviation". Also, does the presence of this channel in an area of (perhaps) relatively low melt variability indicate some seasonal or episodic control on local melt rates?

Fig 4 caption: "The grey lines represents the uncertainty of the difference in basal melt rate." This text seems unnecessary, since error bars are very standard features, but should be "represent" instead of "represents" if this text is kept.

---

## Author Response (AR2)

Dear Reviewer,

We are again grateful for this review! We appreciate your suggestions for improving this manuscript and have taken them into account as far as possible. Please find our answers to your second review in blue colour.

Again, many thanks for your efforts to improve our manuscript!

Best regards,
Ole

**Authors point-to-point response on Referee Comment #2 to tc-2021-230**

I thank the authors for their careful revisions based on the first round of reviews. I still have a few suggestions that should be addressed before publication. These are still mostly to do with Section 3.2, as well as a few other minor comments. Line numbers below refer to the version with tracked changes.

L 95: In my previous review I asked: "What are the possible physical explanations for reasons (1) and (3)?" This question was answered in the authors' response, but please add a brief explanation of these to the text.

> We added the following explanation to the text:
>
> **Line 96:**
> *"An explanation for the low correlation values could be errors in operating the pRES, such as inaccuracies in the alignment of the antennas or incorrectly seated cables. In addition, changes in settings such as the attenuation affected the signal-to-noise ratio, thereby reducing the number of high correlation values."*
>
> In the first round of reviews, we discussed a possibility why correlation values are low around the pore-closure. This is an interesting point as some repeated (A)pRES measurements from Greenland and Antarctica show low correlation values and indicate kind of unrealistic displacements at roughly the depth of the pore-closure. However, the sites that we excluded due to low correlation values that prevent reliable alignment of measurements also have low correlation values over much of the ice column. Hence, we prefer not to discuss this in this manuscript as none of our measurements are a good example of such a feature.

L 105: "The time periods are ranging from 365-9 days to 365-42 days with". This is confusing. Should this be read as "365 minus 9" or "365 to 369", or something else?

> Thanks for your suggestions. We agree that "365 minus 42 days to 365 minus 9 days" may be easier to understand. We have included this in the manuscript.

L 109: should be "same period *as*" rather than "same period *than*"

> Thanks. We corrected this.

L124: Should be "marine ice" instead of "sea ice"

> Corrected.

Section 3.2 is greatly improved. Thank you. I still have some minor suggestions for presenting this section and Figure 4, but I think they can be easily addressed. Below, text from the Authors' Response is in green:

*In this section we want to show the small scale variability of the basal melt rate in order to evaluate the reliability of the large-scale variability. The lower the small scale variability, the more reliable a derived melt rate is for its environment.*

In this case, it seems like standard deviation of melt rate as a function of radius from a point would be a better way to judge the reliability of a point measurement. This would implicitly include the effect of Δhb, without suggesting that Δhb is the primary driver of Δab. However, I am fine with Fig 4 being presented as-is or with some minor changes, with some more explanation in the text (see below)

*Therefore we plotted the deviation of the melt rate against the change in the draft - which is the local ice base slope. This analysis shows that the two nearby stations with the largest difference in the basal melt rate are also those with a large deviation in the draft.*

*The aim of this figure was therefore not to show a trend between the difference in draft and the difference in melt rate. It was all about showing that the variability is generally small and that deviations are connected with changes in the draft.*

However, the data point with the highest draft difference is not obviously a major increase in Δab relative several of those with Δhb < 10m, especially when these uncertainties are taken into account, which makes the effect of draft on melt rate difference unconvincing here. Please present some statistical metric that shows that the Δab at Δhb =14 m is indeed a significant outlier from the values with |Δhb| < 10m.

Also, Δh here takes both positive and negative values. A change of draft of -10m would be expected to have an equal and opposite effect as a change of +10m, but the data point near -10m displays Δa of about 0m. Is there a mechanism to explain a step-change in Δab at |Δhb| = 10m? Perhaps one axis (or both) needs to be a fractional change instead of absolute change?

*The aim of this figure was therefore not to show a trend between the difference in draft and the difference in melt rate. It was all about showing that the variability is generally small and that deviations are connected with changes in the draft.*

*Our intention was to plot the change in melt rate over a quantity that might be of interest and that is accessible. The temperature gradient of the base is unknown. Oceanographic quantities, too, so is a roughness not available. Therefore we have selected the draft here.*

Some text should be added in 3.2 that explains that the choice of Δhb as the independent variable in Fig. 4 is largely made from necessity. Otherwise, the reader assumes that Δhb is presented as the controlling factor on melt-rate, which the sudden increased in Δab at |Δhb| = 10m suggests is not really the case. This sudden increase in Δab is interesting, but it is rather hard to fathom why |Δhb| = 10m would act as a threshold value, and no mechanism is provided for it in the text. Again, the authors do not claim that it is indeed a threshold value, but the choice of Δhb as the independent variable makes it seem this way to the reader.

It also looks like a similar pattern would emerge if the independent variable was ab instead of Δhb. Could it be that higher melt rates (> 1m/yr) also exhibit larger absolute variation (unclear if this is a larger relative variation) because they are driven by processes different from the very low ambient melt rates? It would be helpful to address this possibility in the text.

Thank you very much for this detailed feedback to our Section 3.2 and for your suggestion to replace Fig. 4. Due to the low number of sites, we think showing the standard deviation of the melt rate as a function of radius is maybe not the best way of showing this. However, we think it is a great idea for future measurements to perform measurements on circles with different radius around a camp to be able to derive a standard deviation as a function of distance.

As we haven't done this yet, we would prefer to show the melt rate difference of each site. One way of doing this is, as you have suggested, as a function of distance. Such a figure would support our finding of a low small scale spatial variability. We have included the corresponding figure here.

[Figure]

We agree that this figure may be the best way of presenting the results. However, the former Fig. 4 of the first revised version also included some part of the discussion as it allowed to link the difference in melt rate with a change in draft. We understand your concern that Fig. 4 in the revised version indicates that $\Delta h_b$ is the controlling factor for the melt rate difference. However, it is the only known quantity. You correctly pointed out that at the location with the largest $\Delta h_b$, the difference in melt rate is not exceptionally high (see below), just like at $\Delta h_b$ = -10 m. This weakens the argument that $\Delta h_b$ plays a crucial role here. Since the figure obviously leads to more confusion than clarity, we would like to replace the figure based on your suggestion. As a result, the new Fig. 4 is limited to the results but supports the finding that the small scale variability of the basal melt rate is generally low. By doing this, we hope to keep the text short and clear without having to provide much explanation, which might have been necessary for the former Fig. 4.

Because we replaced Fig. 4, we made some changes in the text: **Line 151 – 155.**

You are right that the melt rate difference of $\Delta a_b$ = 13 m/a at $\Delta h_b$ =14 m is not an outlier based on a statistical metric. By using the definition of outliers from the box-whiskers plot, only those are outliers whose exceed $\Delta a_b$ = 13 m/a. We corrected this in the text (**Line 151**).

We see the point that the former Fig. 4 raises the question if there is a step-change in $\Delta a_b$ at $|\Delta h_b|$ = 10m. In case of a non-linear thermal stratification in the water column, the melt rate may increase sharply beyond a certain draft. Thus, this can be an explanation of a sudden change in melting with deeper draft.

It is true that the two largest $\Delta a_b$ were found at the two largest melt rates. Which is not surprising as a larger $\Delta a_b$ requires one larger and one smaller melt rate. And apart from these two sites, there is no dependency between the two values. Of course, it is possible that the larger melt rates are driven by other processes, but then these vary on short spatial scales. Which leads to the question of why processes differ on these small scales? One explanation could be the change in ice-shelf draft, as the site with the greater depth might have been in contact with warmer waters. Thus, we added a sentence about this in the manuscript:

**Line 163:**
*"Given thermal stratification in the underlying water column, it is also possible for a locally increased draft to result in the ice base having contact with warmer waters, leading to a local increase in melt rates. Since such an increased draft of 10.5 m was found at pRES060 compared to pRES061, this could be a possible explanation for the large difference in melt rate of 0.68 +/- 0.06 m/a."*

L 160: This switches from discussing $\Delta a_b$ to $a_b$, which is not the quantity of interest here.

We are not sure if we correctly linked your comment to the right sentence. The sentence that starts in line 160 is:

*"A change in ice draft of, say, 10m will change the thermal driving by about 0.007°C (e.g. Holland and Jenkins, 1999)."*

*However, this sentence is discussing a change in quantities rather than absolute value. Therefore, it is discussing $\Delta a_b$ and not $a_b$.*

L 165–173: It would help to add a sentence to the end of this paragraph stating that the ice roughness and basal drag coefficient are not known well enough to identify what is in fact driving the variation in melt rates above $\Delta h_b$ = 10 m.

Thanks! We added the following sentence:

**Line 167:**
*"However, neither the ice roughness nor its spatial variation is known well enough to determine its importance in driving local spatial variation in melt rates."*

L 184: It would be helpful to define what counts as "low" variability, as 0.2 m/yr is larger than the value of $\Delta a_b$ at $|\Delta h_b|$ = 14m in Fig 4 that is referred to as a "major deviation". Also, does the presence of this channel in an area of (perhaps) relatively low melt variability indicate some seasonal or episodic control on local melt rates?

You are absolutely right. The inconsistent use of "low" is confusing. As we do not define the site of $\Delta a_b$ = 13 m/a as outlier anymore, this inconsistency is now resolved.

If the presence of the channel indicates some seasonal local melt rate seems to be out of the scope of this manuscript from our point of view. But we are happy to refer you to another manuscript that is currently in review at TC: https://doi.org/10.5194/tc-2021-350

Fig 4 caption: "The grey lines represents the uncertainty of the difference in basal melt rate." This text seems unnecessary, since error bars are very standard features, but should be "represent" instead of "represents" if this text is kept.

Thanks, we removed the sentence!